# Explanation of What Time in Kinematics Is and Dispelling Myths Allegedly Stemming from the Special Theory of Relativity

Roman Szostek 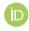

Department of Quantitative Methods, Rzeszów University of Technology, 35-959 Rzeszów, Poland; rszostek@prz.edu.pl

**Abstract:** One of the biggest unsolved problems in physics is explaining what time is. The paper explains what time is in kinematics theories. It has been proved that in the Special Theory of Relativity (STR) and Special Theory of Ether (STE), kinematics time is measured by the light clock. Therefore, all properties of time in kinematics result from the properties of a signal clock. The paper explains the time dilation phenomenon on the basis of STE. The presented explanation is not only a classic description of time dilation but is based on the construction of an innovative technical model of this phenomenon. Time dilation is due to the properties of the light clock. It is a natural property of this clock. The article shows that the claim that the speed of light in a vacuum is the maximum speed in the real world has no theoretical basis. In modern physics, such a doctrine has been adopted as a result of an overinterpretation of the mathematics on which the Special Theory of Relativity is based. The presented model shows how, using atomic clocks, it may be possible to determine the movement in relation to the universal frame of reference in which electromagnetic signals propagate. This article contains only original research.

**Keywords:** light clock; time dilation; time; speed of light; maximum speed

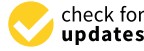



## 1. Introduction

The content of this paper refers to articles [1,2], and, therefore, it is advisable to know them in advance, although it is not necessary. We will rely on four assumptions I-IV that are identical to those of the paper [2]:

I.     Coordinate and time transformation "inertial frame of reference—inertial frame of reference" is linear.

II.    There is at least one inertial frame of reference (called the universal frame of reference) in which the speed of light in a vacuum is the same in each direction. The constant one-way speed of light is indicated by the symbol $c$ = constants.

III.   For each motionless observer in relation to the universal frame of reference, the space has the same properties in each direction, i.e., it is isotropic.

IV.    The average speed of light in the vacuum flowing way back and forth is constant for each observer from the inertial frame of reference. This average speed is indicated by the symbol $c_{av}$ = constants. This average speed does not depend on the observer's speed in relation to the universal frame of reference nor on the direction of light propagation (these results are from the Michelson–Morley and Kennedy–Thorndike experiments).

It can be easily shown that $c_{av} = c$ [1].

Time dilation is a phenomenon manifested by the fact that the duration of the same processes may be different in different reference systems. This means that the duration of the process depends on the speed at which the reference system moves in which the

process takes place. Time dilation is described by kinematics models, in which the passage of time on clocks depends on the speed at which the clocks move.

In classical kinematics, there is no time dilation phenomenon because, in this theory, all clocks measure time in the same way.

In the Special Theory of Relativity (STR), there is no universal frame of reference; thus, inertial systems cannot be attributed to absolute velocities. In STR, the duration of any process measured by the own clock (i.e., motionless clock in relation to the place where the process takes place) is shorter than the duration of this process measured by clocks from other inertial systems (i.e., clocks moving in relation to the place where the process takes place). According to the commonly adopted interpretation of mathematics, on which STR is based, time dilation is relative in this theory, i.e., it depends on which observer measures it. For example, for one observer, process *A* is shorter than process *B*, while for another observer, process *B* is shorter than process *A*. Therefore, in STR, two observers can draw completely different conclusions about the relative duration of two processes if these processes take place in other inertial systems.

In Special Theories of Ether (STE), there is a universal frame of reference in relation to which it is possible to measure the velocities of inertial systems [3–5]. In STE without transverse contraction, motionless clocks in relation to ether are measuring time the fastest. Clocks moving in relation to ether are measuring time slower. The faster clocks move, the slower they measure time. Therefore, in STE without transverse contraction, the time elapse depends on the speed of moving in relation to ether. In STE, all observers evaluate the relative time elapse of any two processes in the same way.

Time dilation in the Special Theory of Ether has different properties than in the Special Theory of Relativity. In the Special Theory of Ether, the speed of physical process (time elapse) depends on the speed in relation to the universal frame of reference at which the inertial system moves, in which the process takes place.

In [2,3], it has been shown that the mathematics on which STR is based can be interpreted differently than it is nowadays accepted in physics. Two other interpretations have been shown. Then STR becomes a theory with a universal frame of reference, i.e., it becomes STE without transverse contraction [2]. With these other interpretations, the time dilation occurring in STR becomes time dilation occurring in STE without transverse contraction. In addition, in the paper [2], it has been shown that the commonly adopted interpretation of STR mathematics is incorrect as it is a theory with desynchronized clocks that cause the unreal time to elapse measurements in inertial systems moving in relation to the observer.

It is believed that time dilation is confirmed experimentally. For example, in the particles of lithium ions accelerated to $c/3$ speed, the frequency of transitions between different energy levels is less than in the same lithium ions that rest motionless in the laboratory [6]. It is concluded that the same processes in accelerated particles are slower than in motionless particles in relation to the laboratory. Another experiment confirming the time dilation was the Hafele and Keating experiment, in which the passage of atomic clock time remains motionless on Earth, and those sent on a trip around the Earth were compared [7].

Indirect evidence for the existence of time dilation is the Michelson–Morley experiment and its improved version, i.e., the Kennedy–Thorndike experiment. In order to explain these experiments within the Special Theory of Relativity and Special Theory of Ether, it was necessary to introduce time dilation into these theories. However, these experiments are not unquestionable proofs of the existence of the time dilation phenomenon because it is possible to explain them without time dilation, with the use of Ritz's emission theory (ballistic theory of light), according to which light has a constant speed only in relation to its source [8].

## 2. Light Clock

### 2.1. Principle of Light Clock Operation

Figure 1 shows a clock, which we will call a signal clock. This clock uses a signal that propagates at a constant speed $v_0$ in a homogeneous medium. The universal frame of reference $U$ is connected with the medium in which the signal propagates. The clock can move relative to the medium in which the signal is propagated (at $v$ speed). The clock is connected to the inertial system $U'$.

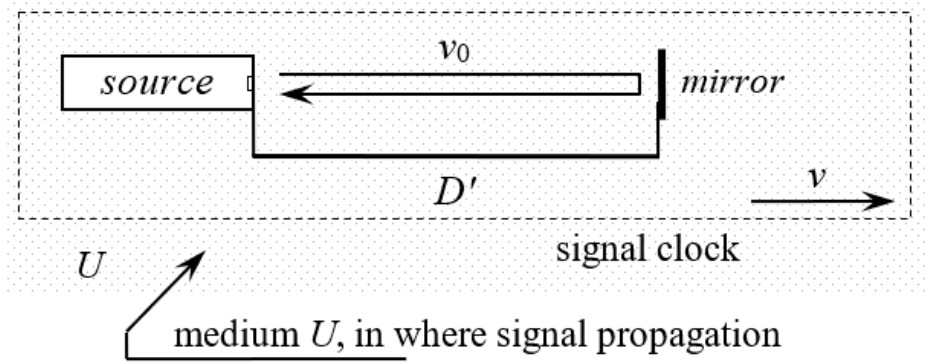

**Figure 1.** The signal clock is based on the signal propagating in the distinguished reference system.

The clock consists of a signal source and a mirror that can reflect such a signal. The signal source and the mirror are rigidly connected at $D'$ distance and form the arm of a clock. The signal is sent from the source and travels a distance of $D'$ length. Then it is reflected by the mirror and returns to the source along the same path. When the signal returns to the source, another signal is sent immediately. This means that the signal clock continuously generates further signals. The total time of one signal passing to the mirror and back is a time standard for an observer in the same inertial system $U'$ in which the clock is located. The duration of one cycle of a signal clock is the smallest unit of time that a signal clock can measure.

If the signal used in a signal clock is a light impulse (or any other electromagnetic impulse), then the clock will be called a light clock. In this paper, we will analyze the properties of a light clock, assuming that light propagates in the universal frame of reference (ether), i.e., according to the Special Theory of Ether [3–5]. On this basis, conclusions will also be drawn from the Special Theory of Relativity.

Every observer measures his time (proper time) with his own light clock, which is motionless in his inertial reference system.

### 2.2. Light Synchronization of Clocks Means Using Light Clocks

The theorem on the signal clock (light clock):

**Theorem 1.** *Assumption: The clocks in inertial systems are synchronized with the signal (light). Conclusion: Clocks measure time exactly as signal clocks (light clocks).*

**Proof of Theorem 1.** The synchronization of two clocks with light is shown in Figure 2.

The two clocks are located in the inertial system $U'$. The distance between clocks is $D' = constans$. When clock A indicates value $t'_{A1}$, a light impulse is sent from it to clock B. If the one-way speed of light in this direction is $c^+$, then when the light impulse reaches the clock B, the following value must be set on it

$$t'_B = t'_{A1} + D'/c^+ \tag{1}$$

Synchronization of clocks must also operate on the opposite side. The light impulse is reflected immediately by clock B and returned to clock A. If the one-way speed of light on

the way back has a value of $c^-$, then when the light impulse returns to clock $A$, it indicates the value

$$t'_{A2} = t'_B + D'/c^- \tag{2}$$

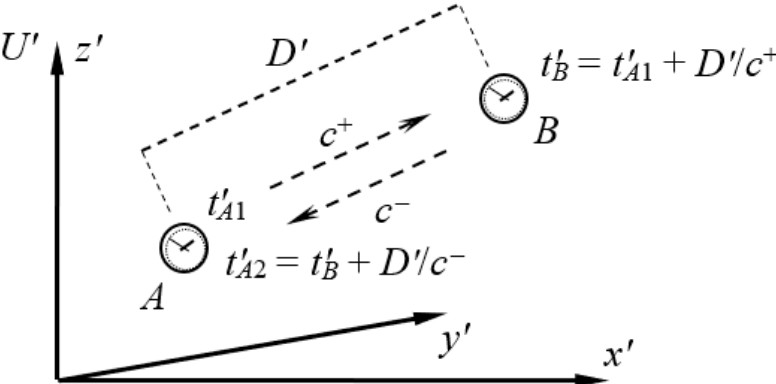

**Figure 2.** Synchronization of clocks with light means that time is measured by light clocks.

On this basis, we can see that the time elapse on the clock $A$ is exactly the same as the total time flow of light back and forth, and it is

$$t'_{A2} - t'_{A1} = D'/c^+ + D'/c^- \tag{3}$$

The time required for the light signal to travel $A{\to}B{\to}A$ is used as a standard for the adjustment of clock $A$. This means that the time required for the light to travel back and forth along $D'$ arm is exactly the same as the time elapse on clock $A$. This means that clock $A$ measures the time in the same way as a light clock. This proves the conclusion. □

Dependence (3) can be transformed into another form

$$t'_{A2} - t'_{A1} = D'\left(\frac{1}{c^+} + \frac{1}{c^-}\right) = \frac{2D'}{\frac{2}{\frac{1}{c^+}+\frac{1}{c^-}}} = \frac{2D'}{c_{av}} = \frac{2D'}{c} \tag{4}$$

The time lapse on clock $A$ in one cycle depends on the arm length along which the light flows and the average speed of light $c_{av}$ on the way back and forth. It is worth noting that the time lapse on clock $A$ does not depend on the one-way speed of light $c^+$ and $c^-$ if $c_{av} = c = constants$ (assumption IV).

Synchronization of $A$ and $B$ clocks depends on the one-way speed of light $c^+$, and therefore synchronization depends on what theory this process is based on (classical kinematics, Special Theory of Relativity, or any of the Special Theory of Ether). However, clock $B$ does not affect the speed at which clock $A$ measures time. Although in the synchronization process the light impulse was sent to clock $B$, the speed at which clock $A$ measures the time depends only on the total time it takes for the light impulse to go back and forth. This means that clock $A$ measures the time exactly like a light clock. Since the requests received for clock $A$ apply to every clock, all light synchronized clocks measure the time exactly the same as a light clock.

Thanks to the claim about a signal clock, it is known that in all kinematics, the standard of time is a signal clock (a light clock). Therefore, all properties of time in kinematics result from the properties of signal clocks.

### 2.3. Time Measurement in Own Clock System (in Inertial Frame of Reference)

The light clock rests in the inertial system $U'$, which moves relative to the universal frame of reference $U$ at speed $v$. We consider the case when the clock arm is parallel to the vector of velocity $v$. For an observer from the $U'$ system, the clock is always a standard

(etalon) of the same time unit. It is the same for an observer from the $U'$ system; the length of clock arm $D'_x$ always has the same value because the clock arm is the same as the standard length.

For an observer with the universal system $U$, the length of the clock arm may depend on the speed at which the clock moves relative to it. We will mark this length with the symbol $D_x(v)$. If $v = 0$, then the length of the clock arm is the same for observers from $U'$ and $U$ systems, and therefore $D'_x = D_x(0)$.

If the clock is motionless in relation to the medium in which the light propagates (universal frame of reference $U$, ether), then the time $t'$, in which the light passes the way to the mirror and back is

$$t' = \frac{2D_x(0)}{c} = \frac{2D'_x}{c} \tag{5}$$

One operation cycle of such a light clock is, for a motionless observer in relation to the clock, a time standard with value (5). It should be noted that if a clock follows one cycle, the motionless observer in relation to that clock always evaluates it as a time elapse of time of the same value (5), regardless of whether they move relative to the universal system $U$ or not.

It is not known what values have a one-way speed of light $c^+$ (when the light moves in the mirror direction) and $c^-$ (when the light moves on its way back to the light source) [1]. As one-way speed $c^+$ and $c^-$ may depend on the direction of light propagation or speed $v$, the one-way light clock, in which the signal flows in only one direction, may not be a good time standard (time etalon). However, if the average speed of light flowing along the way to the mirror and back is constant (assumption IV), then the bi-directional clock is a stable time unit standard. With such a unit, the time will not depend on the direction of light emission, i.e., the way the light clock is set, nor on speed $v$. Therefore, in this paper, we will use a two-way light clock.

### 2.4. Time Measurement from Universal Frame of Reference

Let us consider the situation as shown in Figure 3.

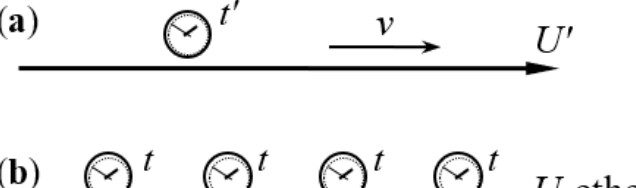

**Figure 3.** Comparison of time elapse: (**a**) clock moving in relation to ether; (**b**) motionless clocks in relation to ether.

We have several identical light clocks. One of these clocks is placed in the inertial system $U'$, while the other clocks are placed in the universal system $U$. The clocks in the universal system $U$ are synchronized with the light, which in the universal system $U$ has a speed of known one-way value $c$ (assumption II). The clock from $U'$ system passes by the clocks from the $U$ system; thus, their indications can be compared. It will be shown below that although all clocks are identical, at the time when the clock from $U'$ system follows one cycle, the clocks from the $U$ system will follow the other number of cycles. This is the phenomenon of time dilation.

For this purpose, we will analyze the distance covered by the light of the clock in the $U'$ system from the observer's perspective in the $U$ system (Figure 4). We consider the case when the arm of the clock is parallel to the vector of velocity $v$ with which the clock moves. Dimensions parallel to velocity $v$ will be marked with index $x$, so in this case, the length of the clock arm is $D_x(0) = D'_x$ in $U'$ and $D_x(v)$ in $U$.

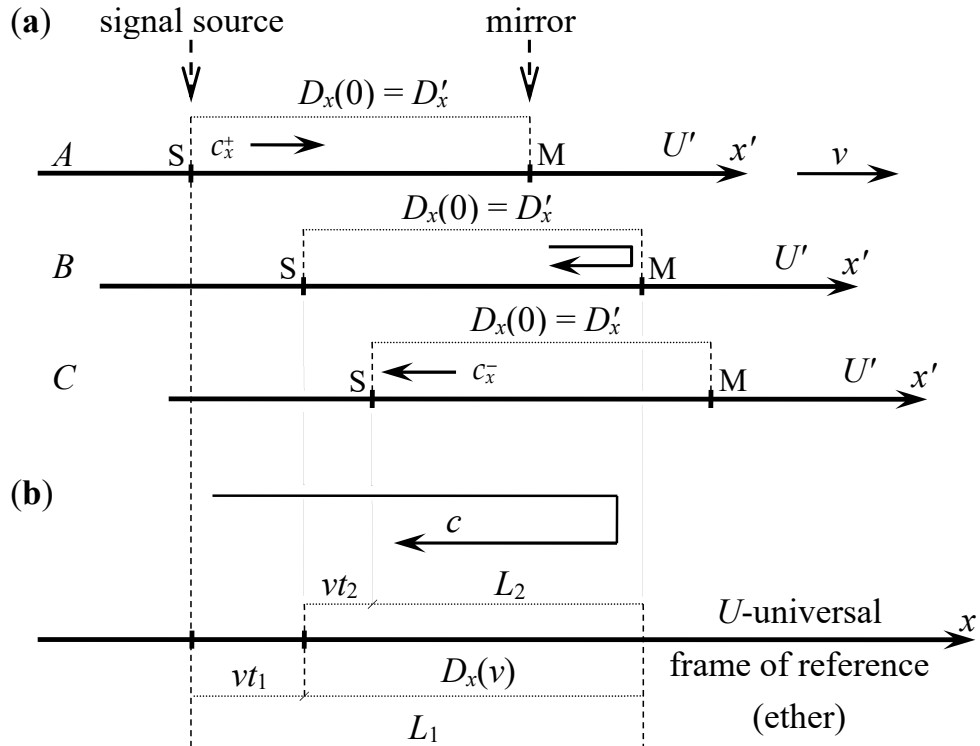

**Figure 4.** Light clock: (**a**) the path of light impulse seen from $U'$ system in which the clock is located; (**b**) the path of light impulse seen from $U$ system in which the light propagates.

Figure 4 in part $A$ shows the moment when a light impulse is emitted from a source (S). Part $B$ shows the moment when the light impulse is reflected by the mirror (M). Part $C$ shows the moment when the light impulse returns to the source (S).

From the perspective of the $U'$ system, the light travels a distance to the mirror of length $D_x(0) = D'_x$ at time $t'_1$, at a speed $c_x^+$. Returning to the source, the light follows a path of the same length $D_x(0) = D'_x$, at time $t'_2$, at a speed $c_x^-$. If the clock follows one cycle, then according to the observer in the same inertial system $U'$, the time elapse $t' = t'_1 + t'_2$ with value (5).

As the dimensions of bodies moving in relation to the universal frame of reference can change, hence from the perspective of the $U$ system, the distance between the light source and the mirror will be marked by $D_x(v)$.

From the perspective of the $U$ system, light travels a distance to a mirror of $L_1$ length, at time $t_1$, at a speed $c$. Returning to the source, the light travels a distance of $L_2$, at time $t_2$, at the same speed $c$. From Figure 4 (part $b$), we obtain

$$L_1 = D_x(v) + v \cdot t_1, \qquad L_2 = D_x(v) - v \cdot t_2 \tag{6}$$

$$t_1 = \frac{L_1}{c} = \frac{D_x(v) + v \cdot t_1}{c}, \qquad t_2 = \frac{L_2}{c} = \frac{D_x(v) - v \cdot t_2}{c} \tag{7}$$

Relations (7) must be resolved in respect of $t_1$ and $t_2$. Then we obtain the time and the path of light flow in the system $U$ in the form of

$$t_1 = \frac{D_x(v)}{c - v}, \qquad t_2 = \frac{D_x(v)}{c + v} \tag{8}$$

$$L_1 = c \cdot t_1 = D_x(v)\frac{c}{c - v}, \qquad L_2 = c \cdot t_2 = D_x(v)\frac{c}{c + v} \tag{9}$$

When the light pulse follows one cycle, then the light signal for the observer from the $U$ system travels a distance of $L_1 + L_2$. Therefore, for the observer from the $U$ system, the time $t$ elapses, which the light needs in this system to cover a distance of $L_1 + L_2$.

$$t = t_1 + t_2 = \frac{L_1}{c} + \frac{L_2}{c} = \frac{D_x(v)}{c - v} + \frac{D_x(v)}{c + v} = D_x(v)\frac{(c + v) + (c - v)}{(c - v)\,(c + v)} = \frac{2\,c\,D_x(v)}{c^2 - v^2} \quad (10)$$

$$t = \frac{2\,D_x(v)}{c}\frac{1}{1 - (v/c)^2} \quad (11)$$

During one cycle of the light clock, for the observer from the $U'$ system, the time elapses given by Equation (5), and for the observer from the universal frame of reference $U$, the time elapses given by Equation (11). Therefore, these observers evaluate the time elapse differently. When we divide the pages of Equation (11) by Equation (5), then we obtain the formula for time dilation in the form of

$$t = t'\frac{D_x(v)}{D_x(0)}\frac{1}{1 - (v/c)^2} = t'\frac{D_x(v)}{D'_x}\frac{1}{1 - (v/c)^2} \quad (12)$$

Time dilation depends on the speed $v$ with which the clock moves (i.e., on the speed of the $U'$ system in relation to the $U$ system) and on how the longitudinal dimensions of bodies moving in relation to the light propagation medium contract (i.e., on the value of $D_x(v)/D'_x$). Time dilation also depends on the speed $c$ with which the signal moves.

From the derived Equation (12), it follows that

$$\frac{\text{number of clock cycles in } U}{\text{number of clock cycles in } U'} = \frac{D_x(v)}{D_x(0)}\frac{1}{1 - (v/c)^2} = \frac{D_x(v)}{D'_x}\frac{1}{1 - (v/c)^2} \quad (13)$$

This means that if the same light clock in $U'$ system is seen by two observers, one is motionless in relation to the clock (inertial system $U'$), and the other is motionless in relation to ether (universal frame of reference $U$), then for them, this clock measures the time differently. It results from the fact that for the observer from the $U'$ system, the considered clock is a time standard, while for the observer from the $U$ system, the time standard is the motionless clocks in his $U$ system. As the speed $v$ with which the clock moves in relation to the universal frame of reference influences the speed of its ticking, the time standard of these two observers works differently.

As a result, time dilation is a natural property of a light clock.

The value $c$ appearing in Equations (12) and (13) does not have to be the speed of light in a vacuum, but it can be the speed of any signal propagating in a medium, e.g., the speed of sound in air or water.

### 2.5. Time Measured by a Moving Clock with a Freely Set Arm

The important question is whether the way the light clock measures time depends on its position in the inertial system $U'$, i.e., on the angle of inclination of the clock arm in relation to the vector of velocity $v$ at which the clock moves in relation to ether. In Sections 2.3 and 2.4, only the case when the clock arm is parallel to the velocity vector $v$ is analyzed (Figure 4). If the way in which the light clock measures time is dependent on the inclination angle, then for the clock to be stable, it cannot be rotated. In such a case, the practical application of a light clock would be difficult.

The experiments of Michelson–Morley and Kennedy–Thorndike essentially consist in comparing the indications of two light clocks, the arms of which are inclined towards each other. The officially recognized results of these experiments show that the average speed of light in a vacuum is always constant in inertial systems available for experiments. This proves that, under real conditions, a light clock measures time independently of the angle of arm inclination. However, if it turned out that Michelson–Morley's and Kennedy–Thorndike's experiments provide some insignificant positive results [9,10], then the way

the time is measured by the light clock would depend to some extent on the inclination angle of this clock in relation to speed $v$.

In papers [1,2], all linear transformations (without revolutions) are derived, for which the average speed of light on the way back and forth is always constant. Each of these transformations is consistent with the zero results of Michelson–Morley's and Kennedy–Thorndike's experiments. In kinematics based on such transformations, the time of light flow along the arm of the light clock (on the way back and forth) does not depend on the inclination angle of this arm to the vector of velocity $v$. Therefore, in these kinematics, the time measured using a light clock does not depend on the position of the light clock in the inertial system. Such kinematics are Special Theory of Relativity and Special Theories of Ether.

In the further part of this paper, we will assume that there is kinematics based on one of the numerous transformations derived from the paper [1]. In each of these kinematics, the light clock measures the time independently of the inclination of the clock arm. Therefore, in all such kinematics, the time dilation of Equations (12) and (13) is valid regardless of the position of the clocks (i.e., regardless of the directions in which their arms are positioned in space).

## 3. Time Dilation in Different Kinematics

### 3.1. Classical Kinematics

In classical kinematics (Galilean), there is no length contraction, and therefore $D(v) = D'$ (for each inclination angle). A light clock without contractions will be called a classic light clock. In this case, the time dilation (12) takes the form of

$$t_{||} = t'_{||} \frac{1}{1 - (v/c)^2} \tag{14}$$

Equation (14) applies only if the clock arm and vector of velocity $v$ are parallel. Therefore, the symbol $t$ is accompanied by the designation $||$.

Initially, it was believed that the universal frame of reference, in which light (ether) propagates, occurs within classical kinematics. For this reason, Michelson and Morley planned their experiment on the basis of predictions resulting from classical kinematics, into which the ether was introduced. Equation (14) shows that if in such a model clocks are synchronized with the help of light, there will be time dilation described by this equation.

As in classical kinematics with ether, the average speed of light depends on the direction of emission (Section 1.3 of [11]); that is why in this theory, the time dilation depends on the inclination of the arm of the light clock in relation to the vector of velocity $v$. It was precisely to detect this effect, although they probably understood it differently, that they counted in their experiment Michelson and Morley.

Time dilation in classical kinematics with ether will not only be when the speed $c = \infty$, or equivalently when $v << c$ because then time dilation is immeasurable. Then Equation (14) is simplified to $t = t'$. Therefore, in the introduction of this paper, it is written that there is no time dilation in classical kinematics.

### 3.2. Kinematics of the Special Theory of Ether

In the paper [1], the whole class of kinematics (transformation) of the Special Theory of Ether was derived, which is in accordance with the zero results of the Michelson–Morley experiment. In these kinematics in every inertial system, the average speed of light is constant (assumption IV is fulfilled). Therefore, the light clocks measure time independently of their inclination in relation to the speed $v$ with which they move in relation to the ether.

Longitudinal contraction (i.e., in the direction parallel to velocity $v$) in STE is expressed by the equation derived from the paper [1] in the form of

$$D_x(v) = D_x(0)\, \psi(v) \sqrt{1 - (v/c)^2} = D'_x\, \psi(v) \sqrt{1 - (v/c)^2} \tag{15}$$

Parameter $\psi(v)$ describes transverse contraction (i.e., perpendicular to velocity $v$), in which the kinematics of the Special Theory of Ether are different from each other. In these kinematics, the transverse contraction is closely related to longitudinal contraction. This is forced by the fact that in every frame of reference, the average speed of light is always constant (assumption IV). The dimensions of the body in motion are shown in Figure 5.

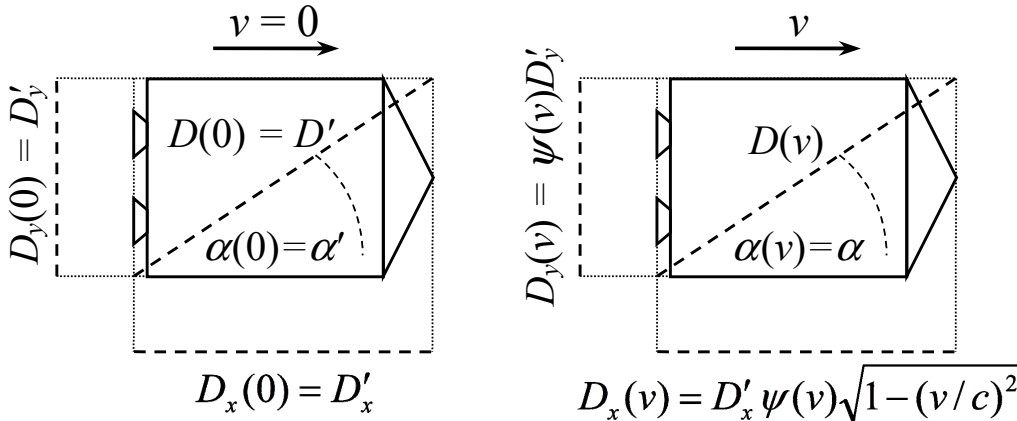

**Figure 5.** Body dimensions in STE observed by a motionless observer in relation to ether.

Based on (15), the dilatation of time (12) takes in STE the form of

$$t = t' \frac{\psi(v)}{\sqrt{1 - (v/c)^2}} \tag{16}$$

Time $t$ is always measured by motionless clocks in relation to the ether, while time $t'$ is measured by motionless clocks in relation to the inertial system moving in the ether at speed $v$. From Equation (16), it follows that each kinematics of the STE has a different time dilation.

The function $\psi(v)$ denotes a spatial contraction. However, this contraction affects light clocks because the arms of the light clock are also affected. When the light clock arm becomes shortened, time passes on it differently. Therefore, indirectly, the function $\psi(v)$ also affects the time contraction (16).

In Special Theory of Ether without transverse contraction $\psi(v) = 1$. In this theory, the time dilation is expressed by the same equation as in the Special Theory of Relativity, i.e.,

$$\psi(v) = 1 \overset{\text{STE}}{\Leftrightarrow} t = t' \frac{1}{\sqrt{1 - (v/c)^2}} \tag{17}$$

There is STE in which there is no time dilation. This is a theory in which transverse contraction is expressed by the equation

$$\psi(v) = \sqrt{1 - (v/c)^2} \tag{18}$$

Then all clocks measure the time in the same way regardless of their motion state. This is due to the fact that on the basis of (18), Equation (16) takes the form of

$$\psi(v) = \sqrt{1 - (v/c)^2} \overset{\text{STE}}{\Leftrightarrow} t = t' \tag{19}$$

Figure 6 shows the time dilation diagrams for the four examples of the Special Theory of Ether.

Now we will determine some additional equations that we will be needed in the next part of the paper.

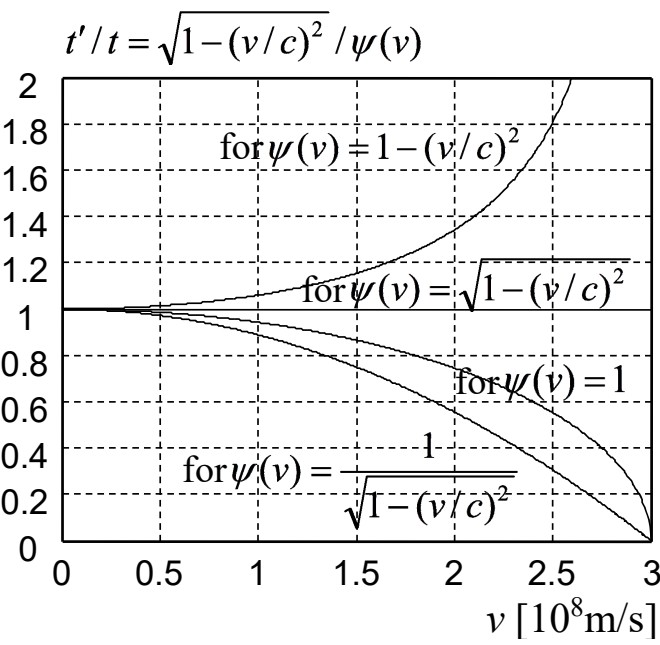

**Figure 6.** Time dilation for four kinematics of the Special Theory of Ether. The graphs show the ratio of time $t'$ measured in the inertial system to time $t$ measured in the universal frame of reference.

Body size measured at any angle to the direction of velocity $v$ is indicated by $D(0) = D'$ and $D(v)$, i.e., without indexes $x$ and $y$. If the angle $\alpha'$ is measured in the body's rest system, then according to the equations given in Figure 5, there is

$$D(v) = \sqrt{D_x^2(v) + D_y^2(v)} = \sqrt{\left(D'_x\,\psi(v)\sqrt{1-(v/c)^2}\right)^2 + \left(D'_y\,\psi(v)\right)^2} \tag{20}$$

$$D(v) = \sqrt{\left(D'\psi(v)\sqrt{1-(v/c)^2}\cos\alpha'\right)^2 + \left(D'\psi(v)\sin\alpha'\right)^2} \tag{21}$$

$$D(v) = \sqrt{\left(D'\psi(v)\cos\alpha'\right)^2(1-(v/c)^2) + \left(D'\psi(v)\sin\alpha'\right)^2} \tag{22}$$

$$D(v) = \sqrt{\left(D'\psi(v)\right)^2 - \left(D'\psi(v)\cos\alpha'\right)^2(v/c)^2} \tag{23}$$

Finally, the dependence for contraction from any angle $\alpha'$ is obtained in the form of

$$D(v) = D'\psi(v)\sqrt{1-(v/c)^2\cos^2\alpha'} \tag{24}$$

As in STE, the bodies in motion are deformed in a different way in the longitudinal and transverse directions (Figure 5), and therefore in inertial systems, the angles are not preserved. Let the angle measured in the resting system of the body as $\alpha'$ has a value $\alpha$ in the ether system. Acting in an asymmetrical way, as in the case of Equations (20)–(24), it is possible to derive the relation to contraction at any angle $\alpha$ in the form of

$$D(v) = \frac{D'\psi(v)\sqrt{1-(v/c)^2}}{\sqrt{1-(v/c)^2\sin^2\alpha}} \tag{25}$$

Based on (24) and (25), the equation is obtained, which binds the angle $\alpha$ and $\alpha'$ in the form of

$$\sqrt{1-(v/c)^2\cos^2\alpha'} = \frac{\sqrt{1-(v/c)^2}}{\sqrt{1-(v/c)^2\sin^2\alpha}} \tag{26}$$

After the transformation of Equation (26), the other equations are obtained, which bind the angle $\alpha$ and $\alpha'$ in the form of

$$\cos^2 \alpha' = \frac{1 - \sin^2 \alpha}{1 - (v/c)^2 \sin^2 \alpha} \ \wedge \ \sin^2 \alpha = \frac{1 - \cos^2 \alpha'}{1 - (v/c)^2 \cos^2 \alpha'} \tag{27}$$

The duration of one light clock cycle is described in Equation (11) by the length of the clock arm, but only when the arm is parallel to the vector of velocity $v$. We will now determine a more general equation for STE, in which the duration of one cycle will be described by the length of arm forming any angle $\alpha$ with the vector of velocity $v$. If the light clock is motionless in relation to the ether, then the length of its arm, regardless of the angle of its inclination, is the same for the observer from the ether. Therefore, for the arms of identical clocks, it is possible to write

$$D(0) = D' = D_x(0) = D_y(0) \tag{28}$$

Equations (28) and (29) apply to a different situation than the one shown in Figure 5. Now the circle is considered, not the rectangle.

If identical light clocks move in relation to the ether at speed $v$, then for an observer from the ether, the arms parallel to velocity $v$ is $D_x(v)$ long and the arm inclined at an angle $\alpha$ to speed $v$ is $D(v)$ long. In STE, these clocks measure time identically, but their arm lengths for an ether observer are not equal in general. In order to determine the relationship between the arm lengths of these two clocks, Equation (25) will be divided by (15), and (28) will be considered. Then the following will be obtained

$$D_x(0) = D(0) \ \Rightarrow \ D_x(v) = D(v)\sqrt{1 - (v/c)^2 \sin^2 \alpha} \tag{29}$$

The searched equation describing the time $t$, measured in the ether system, of one light clock cycle, based on the length of arm $D(v)$ of any inclination, is obtained by taking into account the Equation (29) in Equation (11)

$$t = \frac{2\,D(v)}{c} \frac{\sqrt{1 - (v/c)^2 \sin^2 \alpha}}{1 - (v/c)^2} \tag{30}$$

If to add an Equation (25) to this equation, a different version will be obtained in the form of

$$t = \frac{2\,D(0)}{c} \frac{\psi(v)}{\sqrt{1 - (v/c)^2}} = \frac{2\,D'}{c} \frac{\psi(v)}{\sqrt{1 - (v/c)^2}} \tag{31}$$

Equation (30) is expressed from the length of the moving arm that is measured from the ether system. However, Equation (31) is expressed from the length the arm would have in the ether if it did not move (this is also the length of the moving arm measured in the clock's rest system). It is worth noting that in the second case, in order to calculate the time $t$ of one light clock cycle, it is not necessary to know the angle $\alpha$ or $\alpha'$, but the transverse contraction function $\psi(v)$ is needed.

### 3.3. Kinematics of the Special Theory of Relativity

Different interpretations can be assigned to the mathematics on which STR is based [2]. In modern physics, it is widely believed that time dilation is a property of space-time. This paper shows that it is the property of a light clock. The important thing is the mechanism of the light clock, not the space-time.

The signal clock theorem also applies in STR. It should be noted that the proof of this theorem does not use the reference to ether in any way.

According to the commonly accepted interpretation of STR mathematics, for each observer, the one-way speed of light in a vacuum is constant. Therefore, for each ob-

server, a light clock that is stationary in relation to that observer measures time according to Equation (5), whereas a moving light clock measures time according to Equation (11). Therefore, in STR, equations for time dilation (12) and (13) also apply, but in the same way for each observer. That is, in STR, each observer estimates that a clock (exactly one clock, the same clock at all times [2]) that moves relative to him ticks slower than the stationary clocks in his inertial reference system. It can also be written that in STR, each observer estimates that his own clock measures time slower than clocks that are moving relative to him.

For STR, the time dilation equation should be written in the form of an implication in order to make it clear in which system there is one clock and in which there are many [2]

$$\frac{dx'}{dt'} = 0 \implies dt = dt' \frac{1}{\sqrt{1 - (v/c)^2}} \tag{32}$$

Equation (32) describes the situation where there is one clock in system $U'$ ($dx' = 0$ means that there is only one place to read time in this reference system), and it is compared to many different clocks from system $U$ relative to which it moves.

In the paper [2], it was shown that the mathematics on which STR is based should be interpreted differently. According to this interpretation, clocks in inertial systems are in STR desynchronized. Clocks from different inertial systems are not calibrated correctly, and therefore different observers obtain seemingly contradictory measurements. If the clocks are synchronized (which comes down to the assumption that the parameter $e(v) = 0$), then STR transforms into STE without transverse contraction. Therefore, in STR, the time dilation equation is similar to Equation (17) for STE without transverse contraction.

In STR, the desynchronization of clocks is exactly such that the difference between the inertial system and the universal frame of reference is blurred [2]. The measurements of each observer based on such a desynchronized clock are the same as the measurements of the observer from the universal frame of reference. Therefore every observer, on the basis of their own clocks, determines the dilation according to Equation (32), i.e., as if it was in the universal frame of reference. In this way, all inertial systems become indistinguishable.

## 4. There Is No Reason to Claim That the Speed of Light in a Vacuum Is the Maximum Speed

Based on (16), we conclude that in kinematics, where the following condition is met

$$\lim_{v \to c} \frac{\sqrt{1 - (v/c)^2}}{\psi(v)} = 0 \tag{33}$$

if the time lapse in ether $t < \infty$, then the time lapse in the inertial system has the value

$$\lim_{v \to c} t' = 0 \tag{34}$$

It follows that in kinematics meeting the condition (33), light clocks moving at velocities $v \approx c$ cease to function. If in such kinematics the light clock moves at light velocities or higher, then it does not measure time ($t' = 0$). This is due to the fact that a light impulse moving slower in relation to the ether than the clock cannot catch up with the mirror or source (depending on the direction in which the light clock arm is positioned).

In kinematics, which meets condition (33), it is not possible to describe processes in inertial systems that move at light velocities or higher. In such inertial systems, light clocks do not operate, i.e., it is not possible to measure the time lapse. In the mathematical notation, it is expressed in such a way that when $v \to c$, then peculiarities appear in the time transformations. However, this does not mean that there are any physical reasons for forbidding bodies to reach the speed of light in a vacuum or a speed higher than the speed of light in a vacuum.

Kinematics can be based on light clocks using light impulses that propagate in a medium at a speed of $c_s < c$. This could be performed by a civilization that would live in a

material medium that slows down light, for which vacuum would be unavailable. In their atmosphere, the speed of light would always be, e.g., $c_s = c/2$. In their transformations, there would always be the speed $c_s$, not the speed $c$. Their transformations would stop functioning for inertial systems moving with the speed of $c_s$. If they interpreted it in the same way as the value $c$ is interpreted according to contemporary physics, they would claim that $c_s$ is the maximum speed that cannot be exceeded. This is not the case, of course, and their transformations would stop functioning for the speed of $c_s$ or higher, only because for such velocities, the light clock based on the signal propagating with the speed of $c_s$ does not function.

If the signal clock uses a signal moving at $c_s > c$ speed, then the transformations based on the signal clock will function, also for velocities higher than the light speed in a vacuum. In the extreme case, when $c_s = \infty$, the signal clock functions correctly in all inertial systems, regardless of their speed. Therefore, in this case, the transformations act for inertial systems moving at any high speed. This is the way it is in classical kinematics.

In a similar way, if in classical kinematics, a signal clock using an air propagating sound signal (about 340 m/s) were to be used, then the corresponding transformations would no longer function for velocities such as the speed of sound. However, this does not mean that the speed of sound in the air is an impassable speed.

While in all kinematics that do not meet condition (33), the dimensions of bodies moving at speed $v \to c$ tend for the motionless observer to zero values. This is because

$$\lim_{v \to c} \frac{\sqrt{1 - (v/c)^2}}{\psi(v)} > 0 \;\; \Rightarrow \;\; 0 \le \lim_{v \to c} \psi(v) \le \lim_{v \to c} \sqrt{1 - (v/c)^2} = 0 \tag{35}$$

In kinematics, where condition (33) is not met, the length of a clock arm is contracted due to motion (Figure 5) more than it slows down the classic light clock (Equation (14)). If the classic light clock moves in relation to the ether, then for the motionless observer, the light travels back and forth more slowly. In kinematics that do not meet condition (33), the arm is contracted so much that it compensates for the slowing down effect of a classical light clock. If the clock arm is contracted enough, the clock in motion can measure the time faster than the motionless clock (this is shown in one example in Figure 6, where $t' > t$).

Moreover, in kinematics that do not meet condition (33), the transformations cease to function for inertial systems moving at velocities above $c$. For these kinematics, the reason is that the dimensions of length standards decrease to zero values when $v \to c$ (the arm lengths of light clocks become zero). Therefore, the transformations do not describe the dimensions of bodies moving at velocities $v > c$ and consequently do not describe the operation of light clocks, but this does not mean that the speed of light in a vacuum is an absolute velocity. The theory simply does not say anything about what happens to the dimensions of bodies after passing through a peculiarity when their dimensions are zero.

The analysis presented in this chapter shows that there are no theoretical reasons to argue that the speed of light in a vacuum is impassable. If coordinate and time transformations cease to function in inertial systems moving at velocities of $v = c$ or at higher velocities, it is only because light clocks cannot be time standard in such inertial systems.

Of course, it is not known whether the speed of light in a vacuum is the maximum speed or not. The paper only shows that if the actual speed of light in a vacuum is physically impassable, this is not derived from the Special Theory of Relativity nor from the Special Theory of Ether. The dogma prevailing in modern physics that the speed of light in a vacuum is impassable is that there is currently no theoretical basis for it.

## 5. Atomic Clock, Time Dilation, and Absolute Velocity Determination

As the atomic clocks are subject to time dilation [7,12], it results from the fact that they realize the signal clock. It should be suspected that atomic clocks use a signal that propagates in a distinguished medium (e.g., an electromagnetic signal propagating in the

ether). It is possible that this mechanism is associated with atoms used as oscillators, but it is also possible with other elements of the clock, such as microwave cavities.

There is a large number of atomic clocks, which is why there is no place in this paper to analyze their structure and to search for relations with signal clocks.

Longitudinal and transverse contractions in STE are shown in Figure 5. In the paper [1], it was shown that these contractions must be exactly such if the average speed of light in a vacuum is to have in every inertial system a value *c*.

It is very likely that the actual contractions of the signal clock arms in the atomic clocks are not as ideal as those shown in Figure 5. It is enough for longitudinal contraction to be slightly different or transverse contraction to be slightly different from that shown in Figure 5. Then the timing of such clocks will depend slightly on the way they are inclined in relation to the speed of motion relative to the universal frame of reference. If this effect is very insignificant in our frame of reference, it can be immeasurable using the Michelson–Morley experiment, but it can be measured using atomic clocks.

In order to determine the velocity in relation to the universal frame of reference, an atomic clock should be used, in which the arm of the signal clock contained in it has a strictly defined direction if such a clock can be constructed. For this purpose, several atomic clocks must be placed at different angles to one another on a rotating platform. The rotating platform must be capable of maintaining a constant position of the clocks in relation to space. In order to determine the absolute velocity, it is necessary to look for correlations between the speed at which the clocks operate and the directions in which their arms are positioned in space.

## 6. Coordinate and Time Transformations

We will derive time transformations and position coordinates parameterized by two parameters: longitudinal contraction $\xi(v)$ and transverse contraction $\psi(v)$. These parameters meet the dependencies already given in Equation (15) and in Figure 5, i.e.,

$$D_x(v) = D_x(0)\,\xi(v) \tag{36}$$

$$D_y(v) = D_y(0)\,\psi(v) \tag{37}$$

We accept the markings shown in Figure 7.

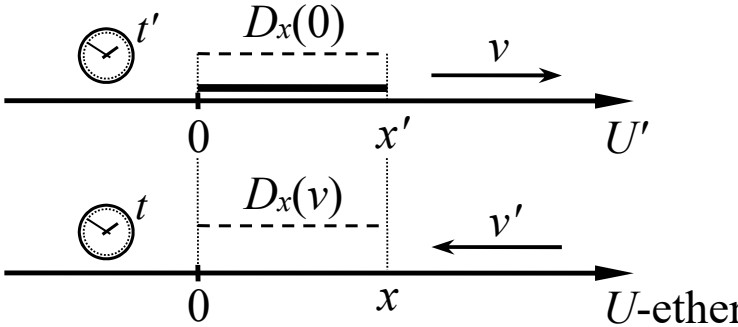

**Figure 7.** Inertial system moves in relation to the ether with speed $v$, while the ether moves in relation to the inertial system with speed $v'$ ($v \cdot v' \leq 0$).

We want the body moving relative to the universal frame of reference to be contracted according to patterns (36) and (37). At the same time, we want the time dilation to be in line with Equation (12) when the arm of the light clock is parallel to velocity $v$. On the basis of (36) dilation (12), we can save it in the form of

$$t_{||} = t'_{||}\frac{\xi(v)}{1 - (v/c)^2} \tag{38}$$

The symbol ∥ indicates that the equation refers to time, measured by clocks, in which arms are parallel to velocity $v$. On the basis of accepted requirements, the following equations are obtained

$$x = \xi(v) \, x' + v \, t_{||} = \xi(v) \, x' + v \frac{\xi(v)}{1 - (v/c)^2} t'_{||} \tag{39}$$

$$x' = \frac{1}{\xi(v)} x + v' \, t'_{||} = \frac{1}{\xi(v)} x + v' \frac{1 - (v/c)^2}{\xi(v)} t_{||} \tag{40}$$

Based on (39), we obtain

$$\xi(v)x' = x - v \frac{\xi(v)}{1 - (v/c)^2} t'_{||} \tag{41}$$

$$x' = \frac{1}{\xi(v)} x - v \frac{1}{1 - (v/c)^2} t'_{||} \tag{42}$$

When comparing (42) and Equation (40), the relation between velocities $v$ and $v'$ is obtained in the form of

$$v' = -v \frac{1}{1 - (v/c)^2} \tag{43}$$

On the basis of (37)–(43), it is possible to save the transformations from the inertial system $U'$ to the ether $U$

$$\begin{cases} t_{||} = \frac{\xi(v)}{1 - (v/c)^2} t'_{||} & \qquad y = \psi(v)y' \\ x = \xi(v) \, x' + v \frac{\xi(v)}{1 - (v/c)^2} t'_{||} & \qquad z = \psi(v)z' \end{cases} \tag{44}$$

The transformation from ether $U$ to inertial system $U'$ has the following form

$$\begin{cases} t'_{||} = \frac{1 - (v/c)^2}{\xi(v)} t_{||} & \qquad y' = \frac{1}{\psi(v)} y \\ x' = \frac{1}{\xi(v)} (x - v \, t_{||}) & \qquad z' = \frac{1}{\psi(v)} z \end{cases} \tag{45}$$

In the paper [2], all linear transformations (without revolutions) have been derived for which the average speed of light in a vacuum on the road back and forth is always $c$. Those transformations are parameterized by two parameters, $e(v)$ and $\psi(v)$. The paper [2] shows that parameter $e(v)$ does not create new kinematics but desynchronizes the clocks in inertial systems. Therefore, all kinematics with a constant average speed of light are parameterized with one transverse contraction $\psi(v)$ parameter. In the paper [1], all such transformations were derived using a different method, based on a geometric analysis of Michelson–Morley's experiment.

Transformations (44) and (45) derived in this paper are more general than those parametrized only by parameter $\psi(v)$. The additional parameter $\xi(v)$ creates new kinematics for which longitudinal contraction can be independent of transverse contraction. In these kinematics, it is possible to model clocks for which the ticking speed depends on the inclination angle of their arms.

Based on the transformation (44) and (45), it can be shown (calculations are omitted) that the average speed of light in the inertial system $U'$ depends on the direction of emission. If the light propagates in a direction parallel to $v$ (i.e., parallel to the $x$-axis), then in the system $U'$, the value of the average speed of light on the road back and forth is

$$c'_{av\_x}(v) = c \tag{46}$$

If the light propagates perpendicular to $v$, then in the system $U'$, the one-way speed of light is

$$c_y'(v) = \frac{\xi(v)}{\psi(v)\sqrt{1 - (v/c)^2}}\, c \tag{47}$$

The average speed of light in each propagation direction has a value $c$ only in those kinematics (44) and (45) where

$$\xi(v) = \psi(v)\sqrt{1 - (v/c)^2} \tag{48}$$

These are kinematics derived from the paper [1].

There are no length contractions in classical kinematics. Therefore, in classical kinematics, we have

$$\xi(v) = \psi(v) = 1 \tag{49}$$

So in classical kinematics, the transformation (44) and (45) takes the form

$$\begin{cases} t_{||} = \frac{1}{1-(v/c)^2}t'_{||} & y = y' \\ x = x' + v\frac{1}{1-(v/c)^2}t'_{||} & z = z' \end{cases} \tag{50}$$

$$\begin{cases} t'_{||} = (1 - (v/c)^2)t_{||} & y' = y \\ x' = x - v\, t_{||} & z' = z \end{cases} \tag{51}$$

Transformation (50) and (51) describe classical kinematics in which time is measured with a light clock, assuming that light propagates in a universal frame of reference at a finite speed $c$. The arms of the clocks are parallel to the speed at which the clocks move relative to the universal frame of reference. If speed $v \ll c$, then transformation (50) and (51) takes the form of Galileo transformation.

## 7. Conclusions

If in the STR and STE kinematics a light signal is used to synchronize the clocks, then the light clock is automatically introduced in these theories as a time standard. In other words, STR and STE are theories in which time is measured by the light clock. These are theories that describe the practical aspects of using such clocks. Therefore, time dilation occurs in these theories.

The paper explains the phenomenon of time dilation. Time dilation is a natural property of the light clock. The paper has shown that the time dilation occurring in the relativistic theories of Special Theory of Relativity (STR) and Special Theories of Ether (STE) is identical to the one that results from the operation of the light clock.

The paper explains why STR transformation (Lorentz transformation) and STE transformations cease to operate when the speed of inertial systems reaches the value $c$ that occurs in these transformations. The presented analysis shows that speed $c$ occurring in transformations is not a speed limit in the physical sense but a speed at which the light clock stops operating. In systems moving at velocities $c$ or above, the light clocks stop measuring time. Therefore, it is not possible to describe the processes taking place in such systems with light clocks. However, from STE and STR results, speed $c$ is physically impassable to exceed.

It has been shown that if classical kinematics is introduced to the measurement of time by means of a signal clock, based on a signal with a finite speed, then also in this theory will appear the time dilation phenomenon. In order for the time dilation phenomenon not to occur, the speed of signal used by the clock must be infinite (or equivalent $v \ll c$). This is how it is in the classical approach to classical kinematics, so there is no time dilatation in it.

The existence of the time dilation phenomenon is indirect proof of the existence of a universal frame of reference (ether) in which the light propagates. If atomic clocks

are subject to this phenomenon, it results from the fact that perhaps they use in their operation a propagating signal in a distinguished medium (it is possible that it is an electromagnetic signal).

Presented analysis shows that one of the ways of testing the movement in relation to ether can be the measurement of time with atomic clocks inclined at different angles to the direction of movement in relation to ether. Perhaps a comparison of time measured with such clocks will enable measurement of the speed in relation to ether if longitudinal or transverse contraction does not compensate for the differences in time dilatation in an ideal way.

The paper [1] shows that Michelson–Morley's and Kennedy–Thorndike's experiments are not able to detect movement with respect to ether in case of an infinite number of different kinematics with ether. In the case of these kinematics, the measurement with atomic clocks will not be effective either. However, if longitudinal or transverse contraction does not compensate for the differences in time dilatation in an ideal way, then the measurement with atomic clocks can be more effective due to the very high accuracy of modern atomic clocks.

Time is what we measure by some kind of physical process. Only cyclic processes, e.g., seasons, pendulum fluctuations (pendulum clocks), quartz crystal vibrations (quartz clocks), atomic oscillations (atomic clocks), or even the reigning periods of successive emperors, are of practical importance. Different physical processes have different sensitivity to changing environmental conditions. Therefore, different ways of measuring time have different individual properties. For example, the time lapse measured by a pendulum clock is sensitive to gravity field intensity, while the time lapse measured by biological cycles is sensitive to different conditions determining life expectancy.

The paper shows that time measurement in kinematics is based on signal clocks and therefore is sensitive to the speed at which the clock moves relative to the medium in which it propagates the signal (STE) or at which the clock moves relative to the observer (STR). In modern relativistic, the time dilation, which is the property of signal clocks, has been automatically transferred to all other physical processes without proper justification. For example, it is not at all certain that biological processes (e.g., the twin paradox) are subject to the same time dilation as light clocks. This should only be verified experimentally. In this way, the idea of time was created, which is a space-time dimension that is not related to any specific physical processes. However, there is no reason to argue with certainty that the speed of the inertial system affects in the same way the frequency of all physical processes, according to time dilation. If the physical process has no relation to the signal clock, the duration of this process may not be subject to time dilation. For example, the life expectancy of a person may depend on the speed of movement in a different way than the speed of ticking (i.e., the period of ticking) of the atomic clock. For this reason, it is not known whether conventional twins, from the paradox of twins, will be subject to time dilation resulting from kinematics. Everything depends on whether the speed of life processes is correlated with the speed of the signal clock. Time dilation refers to signal clocks and physical processes that are related to the signal clock. In the case of other physical processes, the determination of whether they are subject to time dilation requires, in each case, experimental confirmation or some theoretical justification.

From the analysis presented, it follows that time dilation can be interpreted as properties of a light clock and not a property of space-time, as is now believed in the Special Theory of Relativity.

The article showed that the mathematics of STR could be interpreted differently than it is currently performed. According to the presented interpretation, the properties of time in STR result from the properties of the light clock (the theorem on the light clock). Since the light clock stops timing when it is accelerating to the speed $c$ with respect to the observer, therefore the theoretical formulas stop working in inertial systems moving at velocities greater than or equal to $c$. The formulas stop working not because the velocity $c$ is impassable but because the light clocks are not working. It follows from this that the

mistaken belief is that the speed of light in a vacuum is certainly the maximum speed in nature.

So what results from STR mathematics depends on the adopted interpretation of this mathematics. One interpretation (now taken for granted) shows that speed *c* is the maximum speed in nature. However, the interpretation in this article only shows that in inertial systems moving at speed *c*, light clocks do not work. Due to the lack of time measurement, it is not possible to describe the physical processes in these systems.

It should be noted that scientific methods do not allow proving that something is impossible. Something may be impossible under some theory or under how this theory is interpreted at any given time. However, no theory is absolutely certain because perhaps, in the future, a phenomenon will be discovered which will undermine it. Science can only prove that something is possible if it is performed in a practical way. However, the criticism of *c* as a maximum speed presented in this paper is not based on this property of science. The paper shows that even under the Special Theory of Relativity, there are no grounds for treating *c* as a maximum speed. Even if *c* is the maximum speed in nature, it does not follow the Special Theory of Relativity as it is widely believed today. The speed of *c* is only the speed at which the light clock used in the Special Theory of Relativity to measure time stops operating.

The problem that mathematical formulas can be assigned different physical interpretations is not just about the Lorentz transformation. For example, in the article [13], it was shown that gravitational waves should be interpreted as an ordinary modulation of gravitational field intensities.

Time dilation has been the subject of research in numerous publications [14–20].

Numerous works discuss the zero result of the Michelson–Morley experiment, from which time dilation and the Lorentz-Fitzgerald contraction results [21,22]. There are also published papers showing the paradoxes of the Special Theory of Relativity concerning rotating frames of reference [23]. Article [24] investigates the subject of relativistic velocity addition. There are many papers on relativistic mechanics with significant theoretical results. The paper [25] presents the original definition of acceleration in the Special Theory of Relativity, while the paper [26] develops the formalism for three-vector and four-vector relative velocity. The papers [27,28] address important insights into time dilation in relativity, while the paper [29] presents alternative ideas for relativity.

For each kinematics, it is possible to derive many dynamics. Examples of the Special Theory of Ether were derived in the monograph [3]. The examples for the Special Theory of Relativity were derived in the article [30].

**Funding:** This research received no external funding.

**Institutional Review Board Statement:** Not applicable.

**Informed Consent Statement:** Not applicable.

**Conflicts of Interest:** The authors declare no conflict of interest.

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
