# Peer review of "Explanation of What Time in Kinematics Is and Dispelling Myths Allegedly Stemming from the Special Theory of Relativity"

_applsci, doi:10.3390/app12126272_

Round 1
Reviewer 1 Report
The subject of the manuscript is very interesting so I would like the authors of the article to answer the following two important questions:
1- The ψ(v) function in [22] means fold contraction. Is this a temporal or spatial contraction? And why was it added to Lorentz's main equation (14)?
2-Is not the effect of this correction very small?
Author Response
Thank you very much for the Review Report. I took into account all the comments of the Reviewer (attachment).

Reviewer 2 Report
In my opinion some statements have to be revised. Here I consider only that ones reported in the conclusions, but their meanings have to be better explained also in the previous paragraphs of the article.
1)row 649-651, " modern relativistic, the time dilation, which is the property of signal clocks, has been automatically transferred to all other physical processes without proper justification": this is not true.
2)row 655-656, "If the physical process has no relation to the signal clock, the duration of this process may not be subject to time dilation", the author has to better explain the meaning of this statement. In particular he has to describe what he means with "relation to the signal clock" and how it is possible to decide operationally this relation.
3) row 665-666, "From the analysis presented it follows that time dilation is not owned by space-time, as is now believed in the Special Theory of Relativity, only properties of a light clock", space-time is a mathematical construction with its mathematical properties, so the time dilation is a metric properties of the space-time. This properties have their origins from physical observations, so this statement is ambiguous.
4) row 674-685, every physicist knows that c (the light velocity) as maximum speed in nature is not a consequence of the the Special Theory of Relativity (see Einstein, et al).
Author Response

(The authors gave the same response as above.)

Reviewer 3 Report
Dear Editor
This manuscript is based on some main assumptions and the concept of light clock. The author tries to show that the time measured by the time clock, naturally indicates an interesting property, called time dilation. Then, he studies this phenomenon in various kinematics. Also, he discusses that the speed of light in vacuum is not the maximum speed in nature.
As far as I understand, the mathematical analysis is straightforward and competent. The author showed well that the time dilation arises naturally when we use the time clock. But, about the criticism of the light speed as the maximum speed in nature, my opinion is a little different. I think special theory of relativity has well explained the existence of such a maximum speed in nature, though mathematically. In my opinion it is not a dogma as the author claimed in page 13. Although it seems at first as a result of Lorentz transformation so that it leads to imaginary position and time for speeds greater than c, but it becomes stronger when we use the laws of dynamics in STR upon which no material object can be accelerated to a speed greater than c. In fact, I think the author needs to consider the same procedure to make his claim and also needs to consider more mathematical interpretations. On the other hand, special theory of relativity is one of the main pillars of the standard model and even of the most famous theories beyond the standard model.
In summary and unfortunately I recommend against publication of this manuscript.
Regards

Author Response

(The authors gave the same response as above.)

Round 2
Reviewer 1 Report
I think the answers are convincing and the manuscript is acceptable for publication.
Author Response
Thank you very much for the Review Report.

Reviewer 3 Report
Dear Editor
Despite my dissent, the author was able to convince me. Now, I recommend this article for publication.
Regards
Author Response

(The authors gave the same response as above.)
